Medical Imaging with Deep Learning 1–3, 2021

# Scopeformer: n-CNN-ViT Hybrid Model for Intracranial Hemorrhage Classification

**Barhoumi Yassine**                                    BARHOU29@STUDENTS.ROWAN.EDU
**Ghulam Rasool**                                       RASOOL@ROWAN.EDU
*Rowan University, New Jersey, USA*

## Abstract

We propose a feature generator backbone composed of an ensemble of convolutional neural networks (CNNs) to improve the recently emerging Vision Transformer (ViT) models. We tackled the RSNA intracranial hemorrhage classification problem, i.e., identifying various hemorrhage types from computed tomography (CT) slices. We show that by gradually stacking several feature maps extracted using multiple Xception CNNs, we can develop a feature-rich input for the ViT model. Our approach allowed the ViT model to pay attention to relevant features at multiple levels. Moreover, pretraining the "n" CNNs using various paradigms leads to a diverse feature set and further improves the performance of the proposed n-CNN-ViT. We achieved a test accuracy of 98.04% with a weighted logarithmic loss value of 0.0708. The proposed architecture is modular and scalable in both the number of CNNs used for feature extraction and the size of the ViT.

**Keywords:** Computed Tomography slices, Intracranial hemorrhage, CNN, ViT.

## 1. Introduction

Motivated by the recently emerging vision transformer model (Dosovitskiy et al., 2020), we propose a hybrid architecture composed of multiple CNNs for feature extraction and a vision transformer designated for intracranial hemorrhage classification. In our work, we hypothesize that utilizing feature maps extracted from highly crafted CNNs can improve the information content the ViT is processing and the resolution input it attends to. Furthermore, we hypothesize that generating features from the same input image using multiple CNNs leads to a richer feature content with higher resolution. To this end, we used multiple Xception CNN feature extractors, pretrained on separate paradigms using two distinguished datasets.The first CNN model used the ImageNet dataset for pretraining, which was then fine-tuned on the RSNA dataset. The second CNN model was priorly pretrained on the ImageNet dataset and then further pretrained on data generated from ImageNet using a Generative adversarial network (GAN) (Goodfellow et al., 2014) applied on several brain computed tomography images. The idea behind generating the dataset using GAN was motivated by the dissimilarities of ImageNet natural images and the 2D medical images. We reduce these dissimilarities by further pretraining on the generated GAN images to approach a better inductive bias for our target computed tomography dataset.

## 2. Methodology

The Scopeformer model, presented in figure 1, is an extension of the vision transformer (ViT) architecture. It is applied either directly to raw images or to a given "n" number of feature maps extracted from the latest Xception Add layers and concatenated to a single

feature map. We adopted the base ViT variant with 12 encoder layers and a latent vector dimension of 1456. In our experiments, we used the RSNA intracranial hemorrhage dataset (Flanders AE, 2019) by generating 224×224×3 images from the DICOM files (Burduja M, 2020). The input image to each feature extractor is 224×224×3, and the output dimension is 7×7×1024. For multiple CNNs, the size of the input vector will be 7×7×(n1024). A smaller version of n-CNN-ViT models was introduced to reduce the computational complexity of the ViT input, where we use a 1×1 CNN filter after the Xception Add layer to reduce the dimension from 1024 to 128.

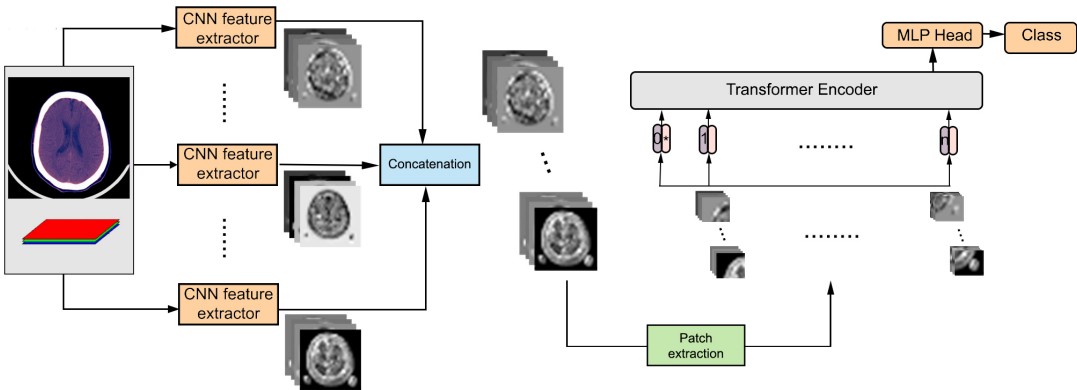

Figure 1: Overview of the proposed n-CNN-ViT architecture. The model is composed of two main stages; Feature map generation and global attention encoding for the MLP head classification.

## 3. Results and discusion

The performances of our models are evaluated by the multi-label weighted mean logarithmic loss (Goodfellow et al., 2014). Figure 2 shows the classification accuracy of our models against the number of CNNs in the feature extractor backbone for different pretraining paradigm settings. We observe that the classification accuracy is directly proportional to the number of Xception models. Moreover, using the same pretrained weights for the 2-CNN-ViT model results in lower accuracies compared to using diverse backbones. Table 1, summarizes best models within each variant. Applying the ViT encoder directly on the raw intracranial hemorrhage images shows that the proposed ViT model by (Dosovitskiy et al., 2020) cannot overcome models without including CNNs as claimed. In fact, The more we add features with a more diverse CNN pretraining paradigms, the richer the feature content will be and the better the ViT attends to the input to extract global attention among patches. The n-CNN-ViT model uses a base ViT variant and relatively small Xception CNN feature maps. Furthermore, We show Comparable results for the smaller version of the 2-CNN-ViT compared to larger ViT inputs. This implies the degree of modularity and scalability of the proposed model.

Table 1: Classification performance of ViT based models on the RSNA validation dataset

| Model | ViT input dimension | Validation accuracy | Loss |
|---|---|---|---|
| **ViT** | 256×256×3 | 94.33% | 0.1822 |
| **1-CNN-ViT (GAN)** | 7×7×1024 | 96.95% | 0.08272 |
| **2-CNN-ViT (ImageNet/GAN)** | 7×7×2048 | 97.46% | 0.07754 |
| **3-CNN-ViT (ImageNet/ImageNet/GAN)** | 7×7×3072 | 98.04% | 0.07050 |
| **2-CNN-ViT (ImageNet/GAN)** | 7x7x256 | 97.58% | 0.07903 |

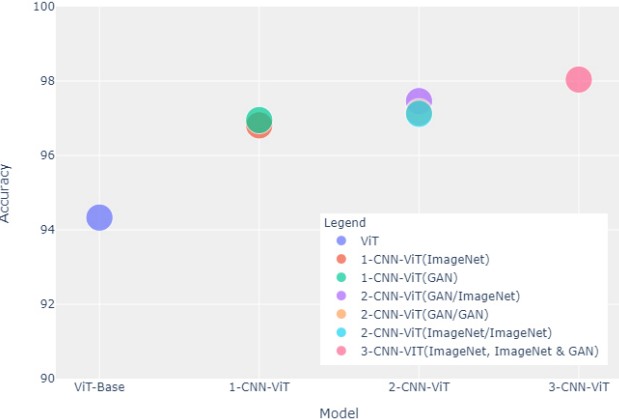

Figure 2: Performance versus n-CNN-ViT variants; Pure ViT, 1-CNN-ViT, 2-CNN-ViT and 3-CNN-ViT, and pretraining modes; ImageNet and data generated using GAN. Models with multiple CNNs and different pretraining modes perform better.

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
