# OpenReview forum: "Scopeformer: n-CNN-ViT hybrid model for Intracranial hemorrhage subtypes classification"
_MIDL.io/2021/Conference/Short — MIDL 2021 Poster_

### Official Review · Reviewer_9x3B · 2021-04-28

**Confidence:** 3
**Final Rating:** 3

**Summary:**

The paper presents a CNN-Transformer hybrid related to the original proposed architecture from the ViT paper. The paper proposes to combine multiple differently pretrained feature extractors to improve the performance of the ViT model and shows that multiple feature extractors improve the performance over a single ViT model.

**Strengths:**

The paper explores the use of a relatively recent method (vision transformers) on medical images. The authors consider different  pretraining settings as well as different dimensionalities for the transformer input. The experiments hint at a better performance of the multi-CNN feature extraction compared to regular ViT or single CNN-features.

**Weaknesses:**

Due to the page limit the paper is very condensed and it is not clear how reproducible the results are and what the exact problem setup is. The usage of GANs for training data generation is not clear and would require further elaboration in a longer paper. It is not clear how the presented approaches compare to regular CNNs and this baseline should be included for completeness and the results seem significantly worse than submissions on the Kaggle leaderboard.

**Deanonymize Review:**

no

**Detailed Comments:**

- It isn't clear what is meant with "[...] and then further pretrained on data generated from ImageNet using a GAN applied on several brain CT images."
- The are some typos like "Furthermore, _W_e show _C_omparable [...]" on page 2.

**Justification Of The Rating:**

I believe the paper could be an interesting discussion ground around how to best apply vision transformers to medical images and move beyond CNN only models. ViT models seem to benefit from huge pretraining datasets or CNNs for feature extraction and it would be interesting to discuss applications in medical imaging.

**Paper Type:**

validation/application paper

**Special Issue:**

no

---

### Official Review · Reviewer_ECXZ · 2021-05-01

**Confidence:** 4
**Final Rating:** 3

**Summary:**

This paper proposes a feature generator backbone composed of an ensemble of convolutional neural
networks (CNNs) to improve the recently emerging Vision Transformer (ViT) models. The authors applied the presented model on the RSNA intracranial hemorrhage classification problem and showed n-CNN-ViT can outperform the ViT model.

**Strengths:**

* The presented idea in this paper is clear

* Relevant related works are properly cited

* The method is built on top of one of the latest developed architectures i.e, vision transformer, and suggests a possible way of improvement over the ViT

**Weaknesses:**


* My main concern is about the validity of the conclusions made by the authors. More precisely, by employing the transformers we know that we are losing some useful inductive biases inherent to CNNs. As a result, the transformer can show its potentials given a large amount of data. However, in the experiments, the n-CNN-ViT is implicitly consuming more data (through transfer learning and GAN & ImageNet data) compared with the simple ViT and as a result, this comparison may not be fair.

Here are the questions:

Does the n-CNN-ViT outperform the simple ViT given the dataset is large enough? To answer this, larger datasets and more experiments are necessary

* The other question is how does a state-of-the-art CNN work on this dataset. Unfortaneoutly there is no comparison against CNNs. If a simple CNN works better than ViT and n-CNN-VIT in a scenario that the dataset is small (which is likely), why do we need to use the more complicated ViT?




**Deanonymize Review:**

no

**Detailed Comments:**

* a space character is missing before "The first...":

"datasets.The first CNN model used the ImageNet..."-------------> "datasets. The first CNN model used the ImageNet..."

* Some part of the method section is presented in the introduction section.


**Justification Of The Rating:**

* I appreciate the idea in the paper.  This paper shows n-CNN-ViT outperforms ViT. However, Dut to the lack of inductive bias in transformers, I think this is likely the better performance of n-CNN-ViT compared with ViT is because of the following two reasons and not having a better architecture.

1- The dataset is small and as a result, the ViT (which has less or no inductive bias) cannot show its potential
2- The n-CNN-ViT is consuming more data (through transfer learning) compared with the ViT

* The authors may argue that employing CNN along with ViT injects some inductive biases and improve the performance of the ViT for the smaller dataset. Although this can be true, the question is if the dataset is small, then what is the advantage of the n-CNN-ViT against a CNN? Unfortunately, there are no experiments for n-CNN-ViT against CNN.

Although this is not clear if n-CNN-ViT always outperforms ViT (especially for larger datasets) or CNNs (for smaller datasets), this paper applies a fairly new model (ViT) to medical images and as a result, I would recommend the acceptance of the paper.

**Paper Type:**

methodological development

**Special Issue:**

no

---

### Meta-Review · Area_Chair_Nx4Y · 2021-05-09

**Recommendation:** Accept (Poster)
**Confidence:** 4

**Metareview:**

Both reviewers find that the paper could stimulate insightful discussions at MIDL and while there are certain shortcomings of this work in progress they are not fundamental. I recommend acceptance.

---

### Decision · Program_Chairs · 2021-05-11

Accept (Poster)